# Research on Hot Stamping-Carbon Partition-Intercritical Annealing Process of Medium Manganese Steel

**DOI:** 10.3390/ma16020576

**Published:** 2023-01-06

**Authors:** Zijian Wang, Xiaoming Guo, Hanlin Ding, Yisheng Zhang, Chongchen Xiang

**Affiliations:** 1School of Iron and Steel, Soochow University, Suzhou 215006, China; 2State Key Lab. of Materials Processing and Die and Mould Technology, Huazhong University of Science and Technology, Wuhan 430074, China

**Keywords:** medium manganese steel, heat treatment process, hot stamping parts, retained austenite

## Abstract

In order to improve the plasticity of hot stamping parts, this paper combines the heat treatment process with the plastic forming of sheet metal, and creatively proposes a new process of hot stamping-carbon partitioning-intercritical annealing. The mechanical properties and microstructure are characterized under the newly proposed process, the quenching-partition (QP) process, and the intercritical annealing (IA) process, respectively. The new process firstly undergoes incomplete austenitizing treatment at 610 °C, then carries out distribution treatment while stamping at 300 °C, and finally conducts annealing treatment in critical zone at 680 °C in two-phase zone. The results show that a multi-phase refined microstructure composed of lath martensite, retained austenite, fresh martensite, and carbides are obtained by the new process. Most of the retained austenite is shaped in the thin film due to martensitic shear, in which carbon and manganese elements diffuse from martensite to austenite by heat treatment, thus stabilizing the retained austenite. Retained austenite with a volume fraction of 33.7% is obtained in the new process. The retained austenite with higher content and better stability is completely consumed during the stretching process, which gives full play to discontinuous TRIP effects, thus delivering the elongation of 36.8% and the product of strength and elongation (PSE) reached as high as 43.6 GPa%.

## 1. Introduction

In recent years, with the increasing attention on resource and environment protection, vehicle lightweight has attracted more and more attention by researchers all over the world [1]. Driven by the trend of automobile lightweight, the application of high-strength steel in vehicles has grown and seen continuous upgrading. At present, advanced high-strength steel has developed to the third generation, in which medium manganese steel is regarded as a typical representative. Excellent mechanical properties can be achieved by proper composition definition and microstructure adjustment in medium manganese steel, which can meet the requirements of vehicle lightweight and crash safety performance as well as low cost. Therefore, this kind of steel has been widely studied [2,3,4].

Due to the high strength and poor deformation ability of high-performance medium manganese steel at room temperature, the deformation resistance, as well as the required stamping force, is larger during the cold stamping process, which easily leads to wrinkles and cracks. In addition, springback deformation always occurs on cold stamping parts, resulting in low dimensional accuracy and other problems [5]. With the aim of solving these problems, hot forming technology has been developed. Hot stamping is a process in which a certain material is heated to the austenitizing temperature, then formed and quenched, and finally, a martensitic structure is obtained. Hot stamping technology is an advanced technology to fulfill automobile lightweight, characterized by lower forming resistance, better formability, higher dimensional accuracy and less springback [6,7]. The tensile strength of the hot stamped parts of medium manganese steel can reach at least 1500 MPa, while their elongation after fracture only ranges from 5% to 10% [8], of which the strength and plastic product is far below the requirements of the third-generation advanced high-strength steel. Therefore, under the premise of ensuring the excellent hot stamping performance of medium manganese steel, it is necessary to explore new forming and heat treatment processes to improve the plasticity to satisfy the performance demands.

Tsuchiyama et al. [9] proposed the interrupted quenching-intercritical annealing process, in which a unique multiphase structure was formed, a large amount of stable retained austenite was obtained, and finally, higher strength and sufficient toughness were achieved. Zou [10] uses two-step intercritical annealing to refine the obtained martensite and retained austenite grains, obtaining excellent low-temperature impact properties. Liu [11] found that the stability of retained austenite can be significantly increased by carrying out QP heat treatment on 5% Mn medium manganese steel, thus effectively improving the elongation of medium manganese steel parts through TRIP effect. The above research shows that the strength and toughness of medium manganese steel can be significantly increased by heat treatment. In order to further improve the properties of medium manganese steel and improve the plastic formability of medium manganese steel, Xu [12] proposed to apply the QPT process and plastic forming technology to TRIP steel, and proposed to integrate the research results of martensitic transformation under stress into the QP process to integrate the plastic forming of casting with the QP heat treatment process. Han et al. [13] carried out hot forming and QP treatment on the commercial hot forming material 22MnB5, which improved the performance of the final parts. Ariza [14,15] introduced the QP process of TRIP800 steel into the hot forming process, and carried out thermal simulation tests at different strain temperatures. Through this new process, it ensured that there was enough residual austenite to produce TRIP effect at the end of forming. From this, we can know that the combination of sheet metal plastic forming and heat treatment process to improve the plasticity of its parts is of great significance to improve the crash safety of formed automobile structures.

On the basis of the research of Tsuchiyama et al., in order to further refine the structure and optimize the properties, a secondary intercritical annealing process was proposed [16,17], but due to the low diffusion rate of the Mn element, longer intercritical annealing time is required. Therefore, this kind of process is not suitable for industrial production applications. Moreover, prolonged annealing will reduce the tempered martensite strength. The QP process has been widely applied in the heat treatment of steel, in which, due to the diffusion rate of the C element being much higher than that of the Mn element, the carbon partition process can be completed at a lower temperature in a shorter time, and the required dual phase microstructure of martensite and austenite was gained [18,19]. Based on the above research, this paper combines the intercritical annealing, QP heat treatment process and sheet plastic forming and proposes a hot stamping-carbon partition-intercritical annealing process. Hot formed parts of medium manganese steel with high strength and ductility can be produced directly by this new process, in which the part performance loss caused by the traditional forming process can be avoided. The paper innovatively proposes an integrated process of heat treatment and hot stamping to achieve more refined grains and better mechanical properties, thereby promoting the application of medium manganese steel in the automotive industry.

## 2. Materials and Methods

The medium Mn steel specimens with a chemical composition of Fe-0.18C-4.6Mn-0.23Si (as shown in Table 1) are vacuum smelted and hot rolled into a plate with a thickness of about 2 mm. In order to determine the phase transformation point of medium manganese steel and formulate the appropriate heat treatment process, the thermal expansion experiment of medium manganese steel was carried out. Figure 1 shows the temperature-dilatometer curve. Using the tangent method, we can see that Ac1 is 595 °C, Ac3 is 740 °C, Ms is 302 °C and Mf is 192 °C. Three kinds of processes are designed as the intercritical annealing, Q&P process and novel hot stamping- carbon partition- intercritical annealing process (shown in Figure 2). The intercritical annealing process is to heat the steel to 850 °C for 10 min for complete austenitization, and then to anneal for 60 min at the two-phase zone temperature (680 °C) between Ac1 and Ac3, of which the sample is abbreviated as IA-680. The Q&P process is to heat the steel to a temperature between Ac1 and Ac3 (close to the Ac1 temperature 610 °C) for 10 min to obtain retained austenite, and then to transfer the steel to a servo press controlled by a thermometer (as shown in Figure 3) for carbon partitioning. The partitioning temperature is 300 °C, the partitioning time is 60 s, and the sample is referred as QP-300. The novel hot stamping- carbon partition -critical area annealing process is to anneal the sample at 680 °C for 30 min based on the QP-300 sample, and the sample prepared by the new process is abbreviated as QA-680.

The tensile strength and elongation after fracture of the sample after heat treatment are measured by tensile test. Dog bone samples (as shown in Figure 4) with a total length of 101.46 mm, a thickness of about 2 mm, a length of 30 mm at the gauge end and a width of 10 mm were used for tensile testing. The tensile experiments are carried out on a WDW-E200 (Instron, Norwood, MA, USA) microcomputer-controlled electronic universal testing machine. Two sets of repeated tensile tests were performed for each process. The microstructure and morphology of the experimental steel in the rolling direction were observed under a SU-5000 (Hitachi, Tokyo, Japan) field emission scanning electron microscope. The prepared mirror-like samples were corroded in 4% nitric acid alcohol solution for 40–50 s. The fine microstructure of the experimental steel, such as martensitic lath and austenite morphology needs to be analyzed by means of electron backscatter diffraction (EBSD). The samples are ground and polished and then electropolished. The electrolytic polishing experiments were conducted under a voltage of 15 V for 10–15 s, in which the solution is the perchloric acid alcohol solution of a 5% volume fraction. The EBSD test multiple is 3000/10,000 times with a test area of 40 μM × 40 μM/8 μM × 8 μM and the test step size of 0.05 μM. The volume fraction of retained austenite was quantitatively calculated by measuring the intensity of diffraction peaks by X-ray diffractometer, with the scanning angle from 50° to 115°and the scanning speed of 1°/min. The sample preparation method was consistent with that of EBSD test. The multiphase structure distribution characteristics and dislocation distribution of the experimental steel were observed by TEM. The TEM samples were ground into thin slices below 100 μM, punched into discs with a φ3 mm puncher and finally thinned by the ion beam. All of the testing and sample preparation methods have been introduced as above for measuring the mechanical properties and microscopic characterization of experimental steels.

## 3. Experimental Results and Analysis

### 3.1. Mechanical Performance

Table 2 shows the mechanical properties of the three processes. As for the three processes, the tensile strengths of QP-300, IA-680 and QA-680 are 1295 MPa, 1027 MPa, 1184 MPa, the elongations after fracture are 20.2%, 30%, 36.8%, and the strength plasticity products are 26.2 GPa%, 30.8 GPa%, 43.6 GPa%, respectively. Figure 5a shows the mechanical properties of three processes. As shown in the graphs of mechanical properties, there exist obvious yield platforms in the stress-strain curves of QP-300 and QA-680 with discontinuous yielding, while IA-680 exhibits consecutive yielding. Sawtooth line phenomena are found on the stress-strain curves of IA-680 and QA-680, which is caused by the TRIP effect of austenite. When a certain critical stress is reached, martensitic transformation occurs on the retained austenite with lower stability, which relieves the local stress concentration and leads to stress relaxation. This phenomenon can also be specifically explained by experiments as follows. Compared with IA and QA processes, the retained austenite content obtained from QP process is much less. Thus, no sawtooth line phenomenon is observed, and the QP-300 sample exhibits higher yield strength and a relatively lower elongation rate. TRIP and yield effects of IA-680 emerge earlier than QA-680, which results from the martensitic transformation beginning under a smaller strain due to the poor stability of retained austenite in IA-680 specimens. In comparison, QA-680 shows higher stability of austenite with a certain amount. Since the mechanical properties of the sample are closely related to the content and stability of retained austenite, excellent mechanical properties are obtained.

The work hardening curves of the three processes are displayed in Figure 5b. The curves can be roughly divided into three stages [20,21]. The first stage is the rapid decline stage, which is mainly associated with the deformation of soft-phase of austenite. In this stage, dislocation slip occurs in austenite which decreases the work hardening rate rapidly. In the second stage, the work hardening rate first increases and then decreases, and the increase may result from the concentrated TRIP effects of the retained austenite with little difference in stability in IA process when it reaches the critical stress. In IA and QA processes, the work hardening rates are featured with serrate fluctuation behavior in the second stage, which is associated with discontinuous TRIP effects. The retained austenite with different stability triggers discontinuous TRIP effects. The serrate fluctuation of IA-680 is larger than that of QA-680, indicating the relatively poorer stability of austenite in IA-680, in which more retained austenite transformation takes place under the strain from 0.025 to 0.15. In comparison, QA-680 is more stable, which provides a lasting TRIP effect and thus contributes to higher elongation. In the third stage, the work hardening rate decreases rapidly, revealing the end of the TRIP effect and the beginning of the necking fracture.

Alloying is conventionally used to improve the mechanical properties of metal materials. However, as the alloying degree increases, the cost of material will increase, the preparation of material will be more difficult and the subsequent weldability will also deteriorate, making it difficult to achieve large-scale application. Nowadays, making material plain is becoming a more and more important index for material development, in which process design plays an important role. The carbon equivalent is usually used to characterize the alloying degree of iron and steel. In this paper, low-alloyed medium manganese steel Fe-0.18C-4.76Mn-0.23Si is adopted. Combing the proposed QA process, the PSE reached 43.6 GPa%, which is higher than most of the reported results [2,9,15,16,17,18,19,20,21,22,23,24] for medium manganese with comparable carbon equivalent, as shown in Figure 6.

### 3.2. Microstructure

Figure 7 depicts the scanning electron microscope images of the three processes. The microstructure of QP-300 mainly consists of tempered martensite (TM), fresh martensite (M’), retained austenite (AR) and carbides. The martensite formed in the first quenching process is tempered at the partition temperature, and finally converted into tempered martensite with an etched shape. During the quenching process, a certain amount of massive austenite will be transformed into fresh martensite creating M/RA island, due to the non-uniform distribution of C element in austenite which is characterized by the low content of C element in the center area and relatively poorer stability of austenite. Silicon element is a carbide precipitation inhibitor. Since the content of silicon added in the experimental steel is not enough, there is still a small amount of carbide precipitation in the microstructure. The microstructure of IA-680 is mainly composed of lath martensite, retained austenite and a small amount of carbides, which mainly presents the interphase morphology of lath martensite and retained austenite. Due to the high temperature of 680 °C, most of the carbides have basically dissolved, with only a small amount of them left. The microstructure of QA-680 is comprised of tempered martensite (TM), fresh martensite (M”), retained austenite (AR) and carbides, which is produced by annealing at the temperature of 680 °C after QP-300 treatment. At this annealing temperature, austenite reverse transformation occurs on the basis of M/RA island and tempered martensite structure. In addition, the generated austenite core area distribution is not uniform, which will be transformed into martensite during quenching, that is, fresh martensite (M”), thus further refining the microstructure.

The bright field TEM images of QP-300, IA-680, and QA-680 are shown in Figure 8. The microstructure of the QP process is mainly composed of martensite, and there is also a small amount of retained austenite distributed on the body matrix. The microstructure of IA-680 and QA-680 are mainly composed of lath martensite and retained austenite. The average widths of martensitic lath for IA-680 and QA-680 are about 165 nm and 140 nm, respectively. In addition, the average widths of austenitic lath for IA-680 and QA-680 are about 152 nm and 128 nm, respectively. The microstructure of QA-680 is more refined than that of IA-680. It can be seen from the TEM images that there are two kinds of retained austenite. One is bulk austenite, mainly distributed in the martensite matrix, and the other is thin-film austenite, extending along the lath boundary.

The equilibrium concentration of manganese in the fcc and bcc phases is different, and it will be partitioned in the two phases during the annealing process. The distributions of manganese element under IA and QA processes are shown in Figure 9a,b, respectively. It can be clearly seen the concentration difference of manganese element in the two phases, the concentration of manganese element in the austenite phase is higher than that in the martensite phase. The mapping results fully illustrate the partition behavior of manganese element during the heat treatments [32]. The line scanning results under QP, IA and QA processes are shown in Figure 9d,f,h, respectively. There are obvious manganese concentration gradients in the austenite and martensite phases and the average weight percentage of different elements can be quantified. The content of manganese element in austenite under IA and QA processes is about 7% and 8.5%, respectively. Higher manganese content means higher stability of the retained austenite, which can explain the higher volume fraction of retained austenite and better mechanical properties of the specimen under QA process. The manganese element is enriched in the austenite region, so the average width of the austenite laths can also be calculated using the line scanning results. The calculated widths of the austenite laths under IA-680 and QA-680 are about 205 nm and 138 nm, respectively, which are consistent with the TEM results. In summary, the samples treated by the QA process have higher austenite volume fraction, better austenite stability and finer structure, resulting in better mechanical properties.

The EBSD diagrams of QP-300, IA-680, and QA-680 are shown in Figure 10 The grey, light blue and yellow area represent the martensite phase, austenite phase and carbide, respectively. Statistical analysis of the results in Figure 10e,f shows that the austenite volume fractions of the IA-680 and QA-680 processes are 18% and 27%, respectively. In addition, the austenite is mainly distributed at the martensitic lath boundaries. However, it is worth noting that the austenite volume fraction obtained by the EBSD method is lower than that measured by the XRD method, as shown in Figure 10. The cause of this phenomenon may be as follows. First, the detection depth of the EBSD method is smaller than the method of XRD, resulting in a greater impact of the surface state on the tested results. Second, the size of some retained austenite is too small to be detected. The austenite reversion during the intercritical annealing can be accelerated by the preexisted austenite [33], which can be obtained by the QP process. By combining the QP and intercritical annealing processes, more retained austenite is stabled and less time is taken.

Furthermore, the presence of the orientation relationship between the retained austenite and parent martensite after IA-680 and QA-680 processes are researched using EBSD method, as shown in Figure 10g,h. Generally, phase transformation mechanisms include the nucleation and growth of new phases by long-range diffusion, short-range diffusion, martensitic or massive transformation, shuffling of atoms and so on. It is well known that the orientation relationship between austenite and martensite satisfies the K-S relationship when the austenite reversion occurs under a diffusional or martensitic transformation mechanism. The K-S orientation is shown by red lines as the interfaces in Figure 10g,h. Most of the boundaries between martensite and austenite coincide with the K-S orientation relationship, which agrees with previous studies that a K-S orientation relationship is mostly observed between martensite and austenite in the lath martensitic steel. During the martensitic transformation, most dislocation structure is inherited from the martensite and more dislocations are generated, resulting in higher dislocation distribution in reversed austenite. As shown in Figure 11, a higher local average misorientation (LAM) value is generated in the austenite phase, which agrees well with the above discussion. The generated strain in the austenite phase can provide additional energy for the martensitic transformation process. As shown in Figure 11c, after the QA-680 process, higher LAM values and more refined LAM distribution are obtained, resulting in the best mechanical properties of the specimens treated with the three processes [34].

The schematic diagrams of the microstructure transformation mechanisms of the three processes are shown in Figure 12. During the partition process, carbon is partitioned from martensite to austenite, and the carbon content near the boundary is higher than in the center. The C-rich region of the retained austenite has a lower martensitic transformation start (Ms) temperature than the room temperature and it can be kept stable in the subsequent cooling. However, when the Ms of the C-lean region is higher than room temperature, the austenite will transform to fresh martensite (M’) during the following cooling, as shown in Figure 12a. Generally, lath austenite and granular austenite are obtained during the IA process and they can be retained through the manganese enrichment, as shown in Figure 1b. As shown in Figure 12c, the prior QP process can introduce preexisting austenite in the initial microstructure before the final IA process in QA-680, which can accelerate the austenite reversion during the final IA process. Moreover, the fresh martensite M’ obtained during the prior QP process acts as the role of refining the mixture microstructure of martensite and austenite.

Figure 13 shows the XRD patterns of the three processes of QP-300, IA-680, QA-680 and the QA-680 sample after fracture. The equation Vγ = 1.4Iγ / (Iα + 1.4Iγ) is applied to calculate the integral intensity of austenite and martensite, and the retained austenite content and carbon content in the retained austenite can be obtained under different processes. For QP-300, IA-680, and QA-680 samples, the retained austenite contents are calculated as 4.01%, 30.7% and 33.7%, and the carbon contents are 1.01%, 0.97% and 1.03%, respectively. Almost no retained austenite peak is found at the fracture of the QA-680 sample, indicating that in the QA-680 sample the retained austenite is consumed during the stretching process and transformed into martensite, which fully proves that the TRIP effect functions on retained austenite during the stretching process. The austenite content and carbon content of the QA-680 are higher than those of the IA-680 process. Moreover, carbon and manganese are the main austenite stabilizing elements in the experimental steel, which diffuse into austenite from martensite in the intercritical annealing process with the diffusion rate of manganese element much lower than that of carbon element. Therefore, it can be inferred that there is a relatively larger content difference of manganese element between both processes, and the QA-680 sample contains more austenite stabilizing elements (carbon and manganese), thus giving rise to better stability of austenite. It is also consistent with the results obtained by the mechanical curves. Furthermore, the QA-680 process delivers more retained austenite with better stability than the IA-680 process based on the mechanical curves, thus triggering the persistent TRIP effects to enhance the strength and plasticity, which results in more excellent tensile strength and elongation rate. Although the IA-680 process can also deliver high content of retained austenite in the previous stage, retained austenite is consumed too early due to its poor stability, which triggers the TRIP effect. As depicted from XRD data, a large amount of retained austenite can be obtained during the intercritical annealing process. However, the stability is poor. In addition, the QP process can deliver a small amount of retained austenite through carbon partitioning with better stability. In comparison, the new process proposed in this paper can produce retained austenite with a higher content as well as better stability by obtaining a certain amount of retained austenite through carbon partitioning and then further increasing the content of retained austenite and enhancing its stability through the reverse transformation of austenite.

## 4. Conclusions

The experimental steel exhibits excellent mechanical properties after the treatment of this new process. The obtained tensile strength is 1184 MPa, the elongation after fracture is 36.8% and the strength-plastic product reaches 43.6 GPa%, which meets the requirement of the third-generation automotive steel. Compared with the QP process and the intercritical annealing process, this newly proposed process obviously enhances the elongation after fracture and contributes to excellent mechanical properties with less difference in tensile strength.Multiphase refined microstructure composed of lath martensite, retained austenite, fresh martensite and carbides are obtained by this new process, in which retained austenite is shaped in film by martensitic shear with its volume fraction of 33.7%. In addition, the retained austenite with better stability and higher content gives full play to the TRIP effect, which facilitates the reduction in stress concentration and the improvement of plasticity, and finally contributes to the excellent mechanical properties.The austenite reversion is accelerated during the IA process because of the preexisting austenite obtained by the prior QP process. Therefore, the volume fraction of austenite under QA-680 is higher than that under IA-680. Moreover, the transformed fresh martensite after QP process can refine the mixture microstructure of martensite and austenite, resulting in a smaller width of the retained austenite laths. The combining of a higher volume fraction of retained austenite and refiner microstructure promotes the improvement of mechanical properties.

## Figures and Tables

**Figure 1 materials-16-00576-f001:**
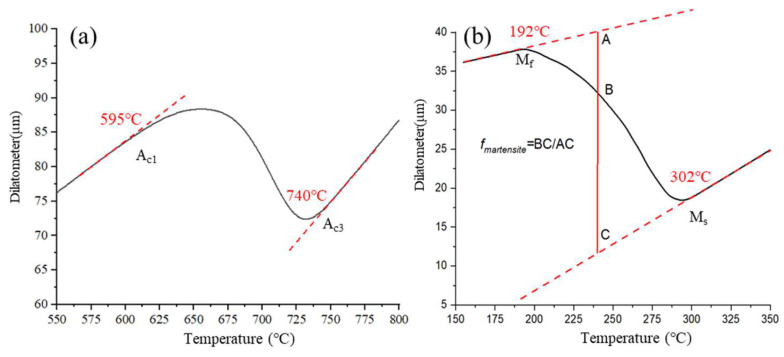
Temperature-Dilatometer curve (**a**) Heating stage; (**b**) Cooling stage.

**Figure 2 materials-16-00576-f002:**
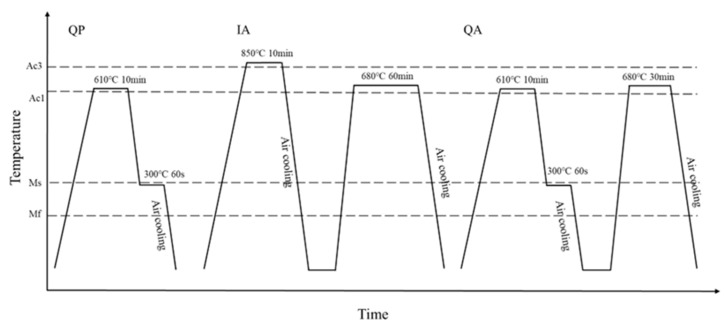
Schematic diagram of QP-300 process, IA-680 process and QA-680 process.

**Figure 3 materials-16-00576-f003:**
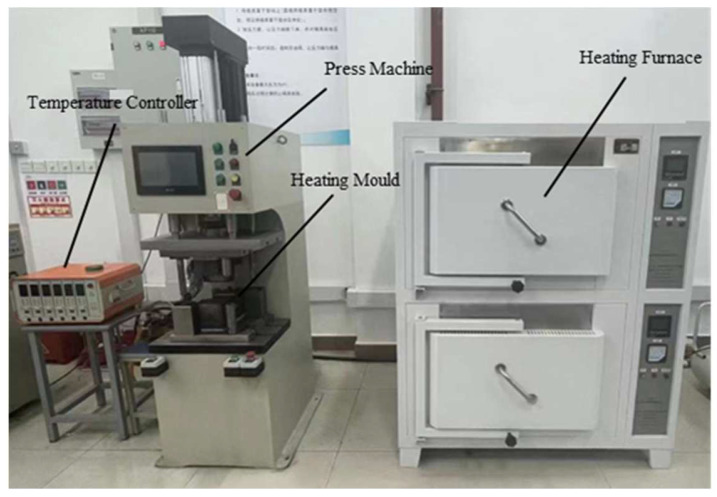
Experimental equipment for all three processes.

**Figure 4 materials-16-00576-f004:**
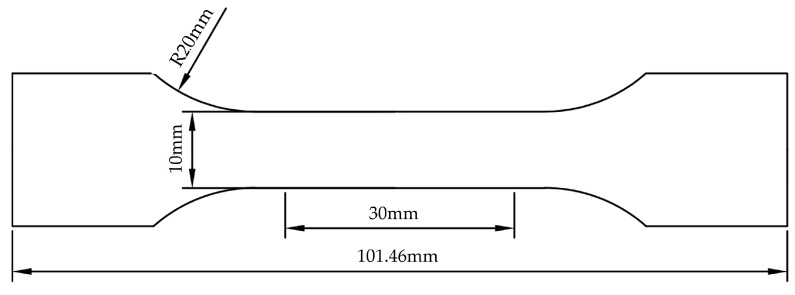
Dimension drawing of dog bone tensile specimen.

**Figure 5 materials-16-00576-f005:**
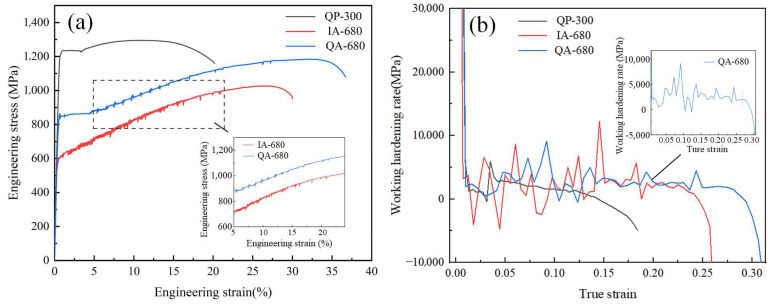
(**a**) Stress-strain curves and (**b**) working hardening rate curves.

**Figure 6 materials-16-00576-f006:**
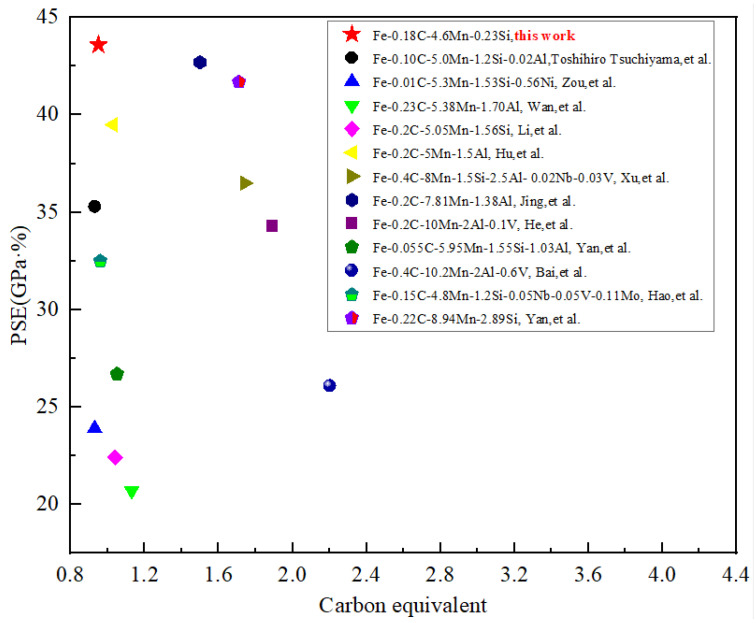
PSE values versus carbon equivalent of this work, compared to reported results [9,10,22,23,24,25,26,27,28,29,30,31].

**Figure 7 materials-16-00576-f007:**
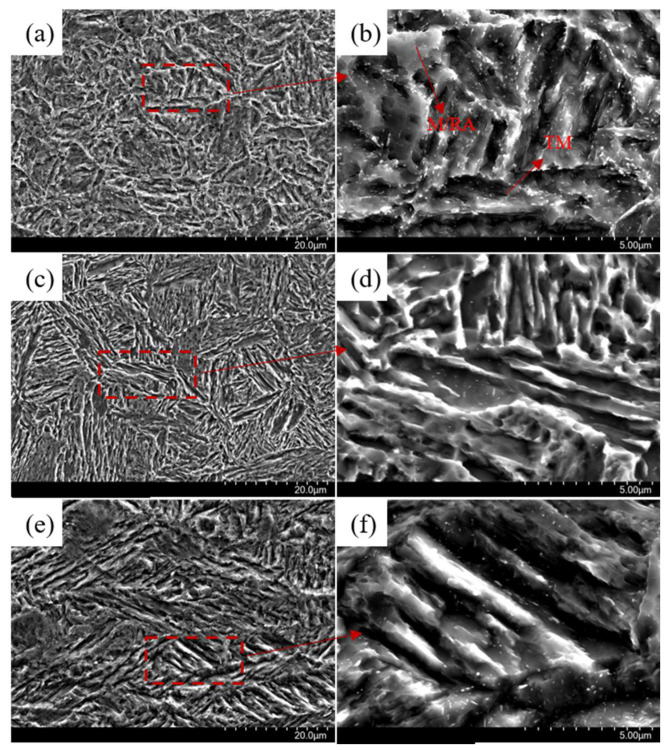
(**a**,**c**,**e**) are SEM images of the three processes of QP-300, IA-680, QA-680; (**b**,**d**,**f**) are the partial enlarged images of (**a**,**c**,**e**).

**Figure 8 materials-16-00576-f008:**
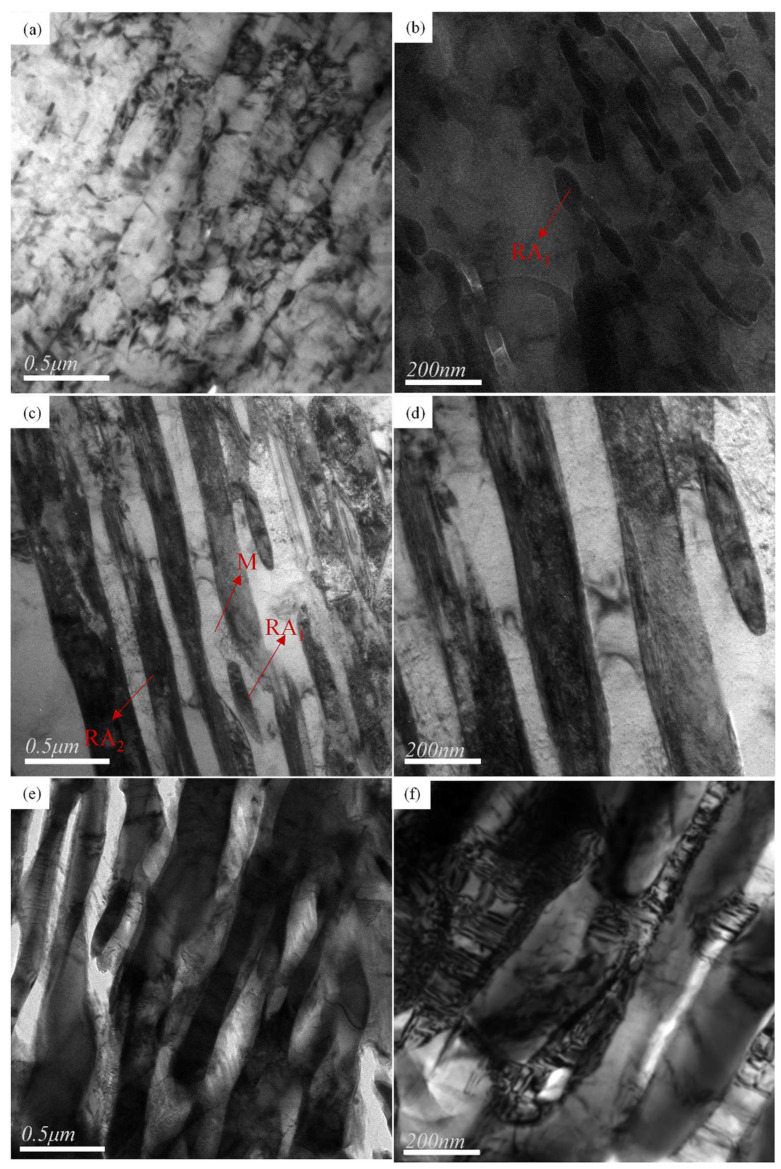
(**a**–**f**) are the transmission electron microscope bright field images of QP-300, IA-680 and QA-680, respectively, in which M is lath martensite, RA1 is granular residual austenite, and RA2 is thin film residual austenite.

**Figure 9 materials-16-00576-f009:**
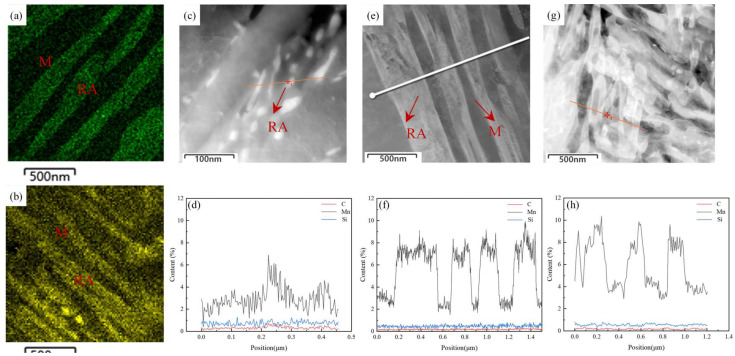
(**a**,**b**) shows the mapping results of the manganese element under IA-680 and QA-680 processes; (**c**,**e**,**g**) are locations of scanning lines under QP-300, IA-680 and QA-680 processes, respectively; (**d**,**f**,**h**) are the line scanning results under QP-300, IA-680 and QA-680 processes, respectively.

**Figure 10 materials-16-00576-f010:**
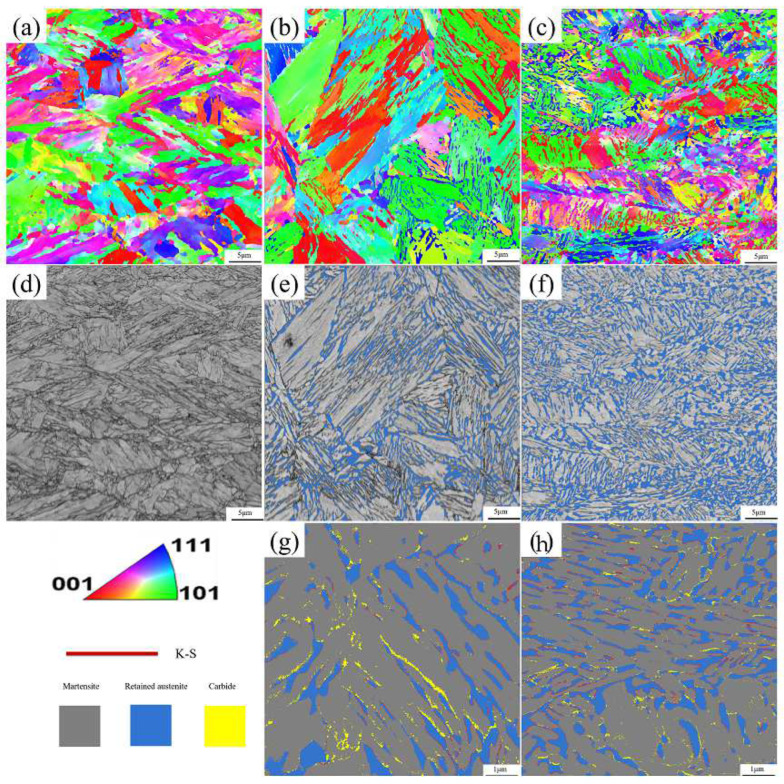
(**a**–**c**) are the IPF maps of QP-300, IA-680 and QA-680 processes; (**d**–**f**) are phase composition diagrams of QP-300, IA-680 and QA-680 processes; (**g**,**h**) are partial enlargements of (**e**,**f**).

**Figure 11 materials-16-00576-f011:**
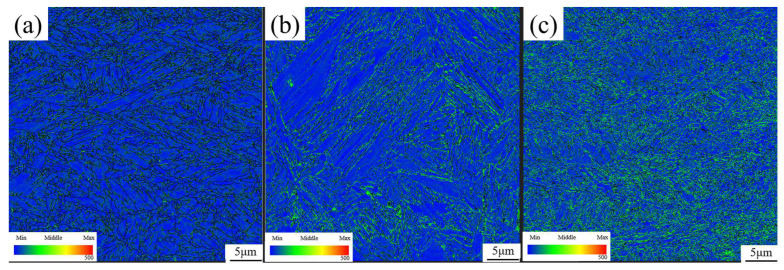
LAM diagrams of (**a**) QP-300, (**b**) IA-680 and (**c**) QA-680 processes.

**Figure 12 materials-16-00576-f012:**
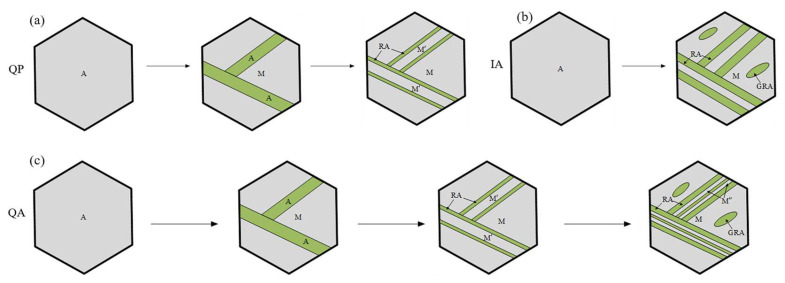
Schematic diagrams of the microstructure transformation mechanisms of (**a**) QP-300, (**b**) IA-680 and (**c**) QA-680 processes.

**Figure 13 materials-16-00576-f013:**
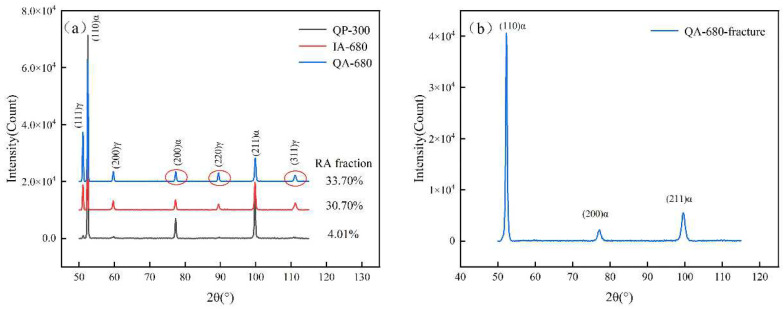
XRD results of (**a**) QP-300, IA-680, QA-680 and (**b**) fractured QA-680 specimens.

**Table 1 materials-16-00576-t001:** The chemical composition (wt.%) of medium manganese steel.

C	Mn	Si	S	P	N	Fe
0.18	4.6	0.23	0.005	0.0182	0.0055	Bal.

**Table 2 materials-16-00576-t002:** The mechanical properties of the three processes.

	UST(MPa)	TEL(%)	PSE (GPa%)
QP-300	1295	20.2	26.2
IA-680	1027	30.0	30.8
QA-680	1184	36.8	43.6

## Data Availability

The data presented in this study are available on request to the corresponding author.

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
