# Peer review of "Research on Hot Stamping-Carbon Partition-Intercritical Annealing Process of Medium Manganese Steel"

_materials, 2023, doi:10.3390/ma16020576_

Round 1

Reviewer 1 Report

- Figure 2 caption: delete "Photo of the";

- Dog bone: ASTM concerning geometry is missing;

- Figure 3: how many tests per sample didi the authors perform? Did the authors consider the repeatability at least of 3 per each sample?

- Figure 6: different scale text style: use the same text style and magnitude. 

Good paper, the only main thing is about the repeatability of the tests, since one test per sample is not the good experimental procedure.

Reviewer 2 Report

The article is a useful study on the hot forming of manganese steels. However, some major revisions need to be made.

My detailed comments were given below.

·         There is no information about process parameters in the abstract section. They should be added.

·         In the introduction section, there is no literature review in detail. There is only 1 paragraph (Page 2 Line 51-62) and it does not contain any depth. It is necessary to add a detailed literature search, including previous studies in the literature and the findings obtained as a result of these studies. In addition, the evaluation of the studies in the literature, and the differences between your own study should be stated and the novelty should be mentioned.

·         How are Ac1 and Ac3 temperature values determined for this steel type?

·         The chemical composition of the steel is not given in detail. It should be given as a table and the elements in its content should be given in detail. Page 2 Although the chemical composition is given in a sentence in Line 72, different elements that affect the Ac1 and Ac3 temperature values are present in the steel. Also, describe your method for determining the chemical composition.

·         The article titled "Microstructural features and mechanical properties of 22MnB5 hot stamping steel in different heat treatment conditions" can be referenced in calculating Ac1 and Ac3 temperatures. Here, information about the calculation of these temperature values is given.

·         The mechanical property values obtained for (QP-300, IA-680, and QA-680) steels should be presented as a table or graphic. This will be helpful to the reader.

·         In the results and discussion section, the results were not supported by the literature. The results given in graphics and figures should be supported by the literature, or if they are inconsistent with the literature, the reasons should be explained.

·         There are many articles on hot forming that have been published in recent years and even next year. However, it has been seen that there are generally old sources in the references section. More new references may be added or some references replaced.

Reviewer 3 Report

The paper should be checked for spelling mistakes.

In my opinion Fig. 4 is not necessary.

Some hardness measurements after each heat treatment should be performed.

References from 15 to 24 are not mentioned in the paper.

Round 2

Reviewer 2 Report

Revisions have been completed and the article has been greatly improved. However, there are 4 elements in the chemical composition of the material given in Table 1. A detailed chemical composition table can be added by performing an EDX analysis.

Author Response

Thanks for the suggestion and the detailed chemical compositions were added to the table 1.